# Analysis of Energy Consumption of the Lyophilizer System Using Solar Absorption Refrigeration

Hong Zhang [1,†], Yun Guo [1,*,†] and Yaolin Lin [2]

1   School of Mechanical and Automotive Engineering, Shanghai University of Engineering Science, 333 Longteng Road, Songjiang District, Shanghai 201620, China; m010120510@sues.edu.cn
2   School of Environment and Architecture, University of Shanghai for Science and Technology, Shanghai 200093, China; yaolinlin@usst.edu.cn
*   Correspondence: graceguo1980@126.com
†   The first two authors contributed equally to this paper and should be considered cofirst authors.

**Abstract:** To design a lyophilizer plant in Guangzhou, the author of this paper used a solar energy absorption refrigeration system and a waste heat of solar absorption refrigeration system. Using Trnsys software, the simulation results show that the solar assurance rate of 74.96% on July 28th can save coal quantity of about 236.8 kg, a solar assurance rate of 53.48% in July can save coal quantity of about 4790.9 kg, and an annual solar assurance rate of 39.06% can save about 40657.1 kg, and each have good environmental benefits.

**Keywords:** lyophilizer; waste heat; solar energy; absorption refrigeration; TRNSYS

## 1. Introduction

The process of vacuum freeze-drying is characterized by subliming the items frozen into a solid state under a vacuum environment, subliming away moisture from the items to eventually dry the items, while sublimating moisture that has been transferred to a cold trap coil to reconstitute into ice [1,2]. This characteristic determines that: the lyophilizer needs to use the heat medium circulation system to heat the material in order to sublimate the water of the article and the ice condensed on the cold trap coil also needs a lot of heat to melt and then discharge.

In the freeze-drying process of the lyophilizer, the cold trap needs to continuously capture the water vapor generated by sublimation so that the cold trap coil will constantly freeze [3]. The ice captured by the cold trap coil needs to melt ice after each lyophilized batch, which in turn requires a large amount of high-temperature defrosting water or defrosting steam. For lyophilizers with a heat medium circulation system, sublimation drying also needs to absorb heat continuously, and the heat required in the whole process is provided by the electric heater of the heat medium circulation system [4]. The preparation of defrosting water or defrosting steam and the electric heater that provides the necessary heat for sublimation need a lot of energy consumption. The excessive energy consumption of the lyophilizer results in a high cost of lyophilization and a serious waste of energy [5]. The heat discharged from the absorber and condenser in the absorption air conditioning refrigeration system needs a cooling tower [6], especially the absorber, whose heat discharged is twice as much as that of the ordinary compression refrigerator [7]. However, the heat after the cooling tower is equivalent to direct waste, and the operation and maintenance of the cooling tower also needs manpower and material resources. How to make full use of the waste heat in the system, effectively reduce the energy consumption of the lyophilizer, and finally achieve the purpose of energy saving and consumption reduction is the research focus of this paper.

## 2. Energy Saving Improvement Strategy of Vacuum Freeze Dryer

Some scholars have done a lot of research on the energy saving of vacuum freeze drying. The analysis of energy consumption in the four stages of pre-treatment, freezing, freeze-drying, and post-packing in the freeze-drying process of 1 ton of spring onions with a 100 m$^2$ freeze-drying machine shows that the energy consumption in the production stage is the largest [8]. Li discussed the measures and means to reduce the energy consumption of a food vacuum freeze dryer from the aspects of raw material pretreatment, a combined drying process, a circulation pressure method, and a controlled freezing point vacuum drying method [9]. Regarding the vacuum freeze-drying during refrigeration and sublimation, running characteristics of each stage, we think through the material pre-treatment and pre-cool temperature, setting a reasonable control of the vacuum freeze-dried room and improving the heating mode so as to optimize the lyophilization process, adopt a combination drying process, improve the design of the lyophilizer, and adopt the methods of strengthening operation management, which can reduce the energy consumption of the lyophilizer and improve the production efficiency [10]. Hu and Pu analyzed the energy consumption composition of vacuum freeze drying and introduced the energy saving measures in the vacuum freeze-drying process from the aspects of the optimization of the freeze-drying process, reasonable use of freeze-drying equipment, selection of a new freeze-drying method, and improvement of freeze drying equipment [11]. Xu and Zhang also discussed the energy saving direction of vacuum freeze-drying equipment from the aspects of the refrigeration system, comprehensive utilization of energy, and application of a solar energy in a freeze-drying machine and vacuum self-freezing technology. Above, they all discussed the energy saving methods and measures of the vacuum freeze dryer from the whole of the vacuum freeze dryer [12]. Shi and Lou took a 20 m$^2$ medical lyophilizer as an example, and on the basis of a comprehensive analysis of the general situation of energy consumption of vacuum freeze-drying equipment, discussed the improvement of energy saving measures regarding the main energy consumption system of a vacuum freeze-drying machine and its refrigeration system [13]. In fact, according to the current situation, the key equipment of lyophilizer at home and abroad has been the most advanced in the world, so it is difficult to save energy from the aspect of this equipment [14].

In the refrigeration system, the heat brought to the system by the power consumed by the pump is ignored [15], while the heat balance Equation (1) of the whole refrigeration device is as follows:

$$Q_h + Q_0 = Q_a + Q_k, \tag{1}$$

In this equation, $Q_h$ is the heat consumption of the generator, kW; $Q_0$ is the refrigerating capacity of refrigeration device, kW; $Q_a$ is the heat load of the absorber, kW; $Q_k$ the is heat load of condenser, kW.

It is easy to see that if the heat discharged from the absorber and condenser is not recycled, the energy consumption of other heat media systems is needed to provide heat for the refrigeration system, which seriously causes a waste of energy. In other words, the research and design of energy-saving and consumption reduction should be carried out from the waste heat of the refrigeration system.

## 3. Utilization of Waste Heat in the Refrigeration System of a Vacuum Freeze-Dryer

### 3.1. Utilization Status of Condensation Heat in the Refrigeration System of a Vacuum Freeze-Dryer

The vacuum freeze dryer must heat the frozen material in the sublimation drying stage to sublimate and escape the frozen ice crystals in the material [16]. At present, most heating systems adopt electric heating or steam heating. Among them, the electric heating mode is the preferred heating mode of the vacuum freeze-dryer because of its advantages of safety, cleaning, convenient control, and simple equipment. At the same time, the sublimation drying time accounts for more than 80% of the total time in the whole operation process of the lyophilizer [17]. Whether it is electric heating or steam heating, the cost is much lower.

While heating the materials in the freeze-drying box, it is necessary to use the refrigeration unit to cool the cold trap of the freeze-drying machine at a low temperature, so as to freeze the water vapor generated in the heating sublimation process, maintain the vacuum in the freeze-drying machine, and maintain the vacuum conditions for the continuous sublimation of ice crystals [18,19]. While the refrigeration unit cools the cold trap, it also needs to discharge the heat of the refrigeration unit to the outside through the condenser and absorber. Therefore, the waste heat discharged by the refrigeration unit can be considered as the heat source of the freeze-drying box.

The comprehensive utilization of heat discharged from the refrigeration system was put forward and put into practice as early as 20 years ago. However, due to the inconvenient adjustment of exhaust heating at that time, and most of the sublimation time, the exhaust heating of the compressor of the refrigeration system is greater than the heat required for sublimation. At this time, it is difficult to use the exhaust of the compressor to heat, resulting in poor economy of compressor exhaust heating [20]. To solve this problem, Lin Yongjin et al. proposed to add an electric three-way valve at the outlet of the compressor and adjust the opening of the electric three-way valve according to the outlet temperature of the refrigerant of the exhaust heater to adjust the amount of exhaust heating, but this method can reduce the efficiency of the compressor. It is easy to lead to the instability of the temperature of the refrigerant, and then a water source heat pump heating mode with cooling water as the low-temperature heat source is proposed [21]. However, this method will increase the number and complexity of lyophilizer equipment.

### 3.2. The Utility Model Relates to an Energy Saving System for a Lyophilization Machine Using Solar Energy Absorption Refrigeration

This paper presents an energy saving system of the lyophilizer using solar energy absorption refrigeration. This energy saving system has been granted a Chinese patent (national patent number: ZL201810620783.1), and the schematic diagram of the system is shown in Figure 1.

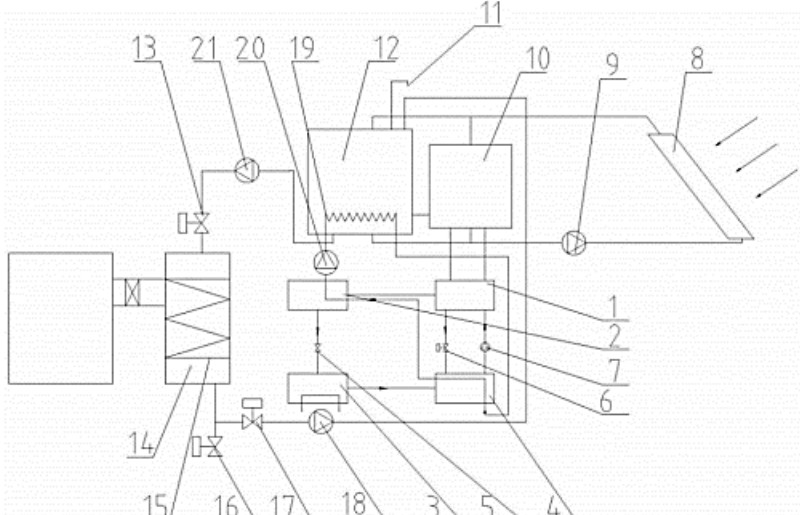

**Figure 1.** Energy saving system diagram of a lyophilizer using solar energy absorption refrigeration. 1—generator; 2—condenser; 3—evaporator; 4—absorber; 5—refrigerant throttle valve; 6—solution throttle valve; 7—solution pump; 8—solar collector; 9—solar energy heat collecting circulating pump; 10—refrigeration hot water tank; 11—water tank overflow port; 12—defrost storage tank; 13—defrost valve; 14—cold trap of lyophilizer; 15—cold trap coil; 16—cold trap discharge valve; 17—defrost water recovery valve; 18—defrost water recovery pump; 19—air conditioning heat exchanger; 20—air conditioning waste heat recovery pump; 21—waste heat using circulating pump.

### 3.2.1. Composition of the System

The invention relates to an energy saving system of a lyophilizer using solar energy absorption refrigeration. The system consists of a total of three units, which are the solar energy collection unit, absorption refrigeration unit, and heat storage unit. Each unit is composed of its own components, and each unit contains only the components that play a major role in the operation of the system, and the rest of the components are not shown in the system. The specific composition of each unit is as follows: Absorption refrigeration unit is composed of generator 1, condenser 2, evaporator 3, and absorber 4. The solar heat collecting unit comprises a solar collector 8, a refrigeration hot water tank 10 connected with the solar collector, and an electric heater for heating the refrigeration hot water tank. The heat storage unit comprises a defrost storage tank 12 and an air conditioning waste heat recovery mechanism.

In the heat collection system, the solar collector, as the heat source of the system, forms a closed heat collection loop with the heat storage tank and circulating water pump, etc [22], and collects heat for the system throughout the year. Therefore, it is particularly important to improve the efficiency of the collector for saving energy consumption. The system adopts the vacuum tube collector, which is usually composed of side-by-side transparent coaxial glass tubes. The vacuum state is formed between the two glass tubes, which eliminates the convection heat loss and reduces the heat conduction loss. The vacuum tube has a large heat capacity, which can obtain heat in a short time. Additionally, the heat pipe has the characteristics of one-way heat transfer, so that hot water will not dissipate downward to the surrounding environment along the heat pipe at night. The vacuum insulation effect is better than that of a flat plate [23]. In addition, vacuum tubes have excellent frost resistance and can be widely used in many fields.

### 3.2.2. The Connection Mode of Each Component

The connection mode of each component is as follows: generator 1, condenser 2, evaporator 3, and absorber 4 are successively connected through the refrigerant pipeline, and refrigerant throttle valve 5 is arranged on the refrigerant pipeline between condenser 2 and evaporator 3. A solution circulation pipeline is arranged between the generator 1 and the absorber 4, and a solution throttle valve 6 is arranged on the solution circulation pipe between the generator 1 and absorber 4, and a solution pump 7 is arranged on the solution circulation pipe between absorber 4 and generator 1, along the direction of medium flow.

The defrost storage tank 12 is connected with solar collector 8 and the refrigeration hot water tank 10. The solar heat collection circulation circuit is provided with a solar heat collection circulation pump 9. The refrigeration and hot water tank are connected to generators 1 and 10 for heat exchange, defrosting tank 12 by air conditioning waste heat recovery, and the absorption refrigerating unit connected to the condenser and absorbers 2 and 4, used as a condenser, and absorbers 2 and 4, used to recycle waste heat, while air conditioning pumps 20 sets of recovery of waste heat in air conditioning waste heat recycling on the pipeline. The defrost water circulation pipeline is connected between the defrost storage tank 12 and the cold trap 14 of the lyophilizer, and the top of the defrost storage tank 12 is provided with a water tank overflow port 11. The air conditioning waste heat exchanger 19 is arranged inside the defrost storage tank 12, and the two are communicated with each other and are connected with the air conditioning waste heat recovery pump 20. The cold well 14 of the lyophilizer is provided with a cold trap coil 15 and a cold trap discharge valve 16. For the defrost water circulation pipeline in this example, along the direction of medium flow, the defrost water circulation pipeline between the defrost storage tank 12 and the cold trap 14 of the lyophilizer is provided with a waste heat utilization circulation pump 21 and defrost valve 13. Defrost water recovery valve 17 and defrost water recovery pump 18 are arranged on the defrost water circulation pipe between the cold trap 14 of the lyophilizer and the defrost water storage tank 12.

The system adopts ammonia absorption refrigerator, which uses $H_2O$ as an absorbent and $NH_3$ as a refrigerant, uses the heat provided by solar energy as compensation and uses

the characteristics of solution to cool. The ammonia absorption refrigerator can produce low temperature below 0 °C [24], and the refrigerant pair will not crystallize, making it is easier to realize air cooling. Its working principle is as follows: The collector converts the radiant energy of the sun into heat energy to heat the circulating water in the collector, and the circulating hot water is used as the heating heat source of the generator to heat and evaporate the ammonia in the solution into ammonia; ammonia enters the condenser and is cooled into liquid ammonia by the cooling water in the condenser pipe; the refrigerant liquid under the condensation pressure is depressurized through the throttling device and enters the evaporator to absorb heat and evaporate into ammonia for refrigeration; the diluted ammonia solution generated by the heated evaporation of the generator flows back to the absorber through the solution heat exchanger to absorb the ammonia generated by the evaporator, enters the solution heat exchanger through the solution pump, and enters the absorber again to complete the circulation. The system diagram is shown in Figure 2.

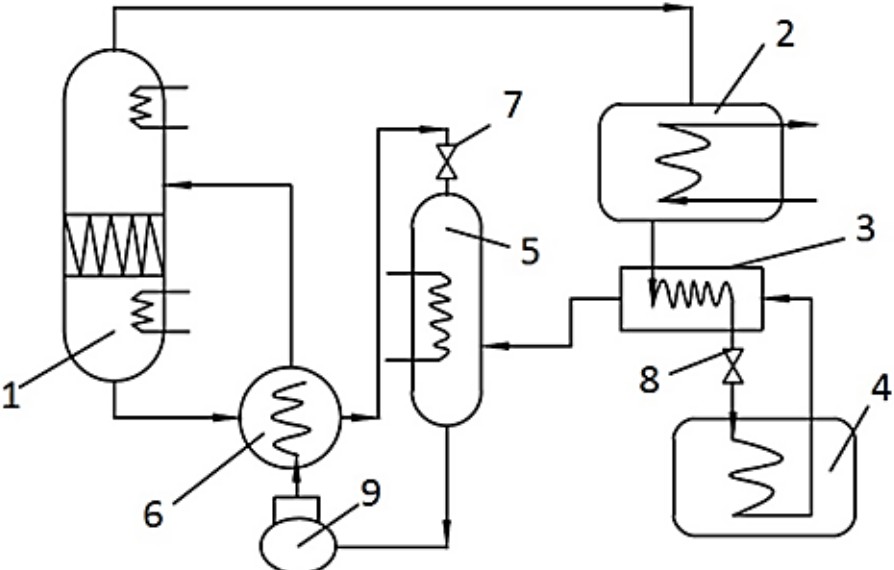

**Figure 2.** Ammonia absorption refrigeration system. 1—distillation tower; 2—condenser; 3—supercooler; 4—evaporator; 5—absorber; 6—solution heat exchanger; 7, 8—throttle valve; 9—ammonia pump.

### 3.2.3. System Operation Scheme

The specific operation scheme of the system is as follows:

1. When the absorption refrigeration unit is working, the waste heat generated by condenser 2 and absorbent 4 is connected to the waste heat exchanger 19 of the air conditioning in the defrost storage tank 12 through the waste heat recovery pump 20, and the waste heat of the air conditioning is used as the heat source to heat the water in defrost storage tank 12. The hot water prepared in defrost storage tank 12 can promote the heat collection cycle of refrigeration hot water tank 10 and solar collector 8. There are two states according to whether the cold trap 14 of the lyophilizer needs defrosting:

   - If the lyophilizer is not running, the defrosting valve 13 is closed at this time. The heated water in the defrosting water storage tank 12 circulates between the defrosting water storage tank 12 and the solar collector 8 under the action of the solar heat collection circulating pump 9. At the same time, it exchanges heat with the hot water cooled by the generator 1 in the cooling hot water tank 10 to reduce the load of the solar collector 8.
   - When the lyophilizer is in operation, frost will form in the cold trap 14 of the lyophilizer, which needs to be defrosted. At this time, the defrost valve 13 is

opened. At this time, the moisture of 12 in the defrost storage tank exits a branch, and under the action of the waste heat circulation pump 21, it enters the cold trap 14 of the lyophilizer through the defrost valve 13, and the high temperature defrost water removes the frosting on the coil 15 of the cold trap. At this point, the defrosting water recovery valve 17 opens, and the melted low-temperature defrosting water enters the defrosting storage tank 12 again under the action of the defrosting water recovery pump and is heated by the waste heat exchanger 19 of the air conditioning with the waste heat of the air conditioning, thus forming a defrosting water cycle. Because defrost storage tank 12 is connected with refrigeration hot water tank 10 and solar energy collector 8 by solar energy heat collection circulation pump 9, defrost water can play a good defrost effect under the heating of double heat sources. When the defrost is finished, defrost valve 13 and defrost water recovery valve 17 are closed. At this point, the water in the defrost storage tank 12 will circulate in the solar heat collection cycle circuit again, and the auxiliary heating cycle will be carried out on the refrigeration hot water tank 10.

2.　When the absorption refrigeration unit stops working, the solar heat collection unit heats the defrosted water in the defrost storage tank 12, and the electric heating of the solar heat collection unit itself can also meet the heat required by the defrost storage tank 12. If it is necessary to defrost the cold trap 14 of the lyophilizer, the defrost water circulation pipeline is opened and the defrosted water is used to defrost.

## 4. System Energy Saving Calculation

### 4.1. Establishment of Trnsys Simulation Model

In this paper, a lyophilizer factory in Guangzhou, China is used as the research object, using TRNSYS (Transient System Simulation Program) software lyophilizer workshop for this system to carry out the simulation analysis of energy saving. The freeze-drying machine plant covers an area of about 1176.6 m$^2$, with a total construction area of 6120 m$^2$. In order to facilitate TRNSYS model building and load calculation, the lyophilizer unit is simplified into a room with an area of 75 m$^2$. The relevant parameters set by the system are shown in Tables 1 and 2.

**Table 1.** Relevant parameters of absorption refrigerator.

| Parameter Name | Unit | Value |
|---|---|---|
| condensation pressure | kpa | 10.09 |
| evaporation pressure | kpa | 0.87 |
| cooling water inlet temperature | °C | 32 |
| chilled water inlet temperature | °C | 12 |
| chilled water outlet temperature | °C | 7 |
| absorber cooling water outlet temperature | °C | 37 |
| condenser cooling water outlet temperature | °C | 41 |

**Table 2.** The parameters of solar collector and water tank.

| Parameter Name | Unit | Value |
|---|---|---|
| Solar system design inlet water temperature | °C | 20 |
| solar system design outlet water temperature | °C | 60 |
| collector area | m | 411.9 |
| collector inclination angle | ° | 22 |
| water tank heat loss | W/(m$^2$·K) | 0.4 |
| water tank capacity | m$^3$ | 30 |
| water tank height | m | 3 |
| Pump power | kW | 7.5 |

The simulation model of the lyophilizer system using solar absorption refrigeration is shown in Figure 3.

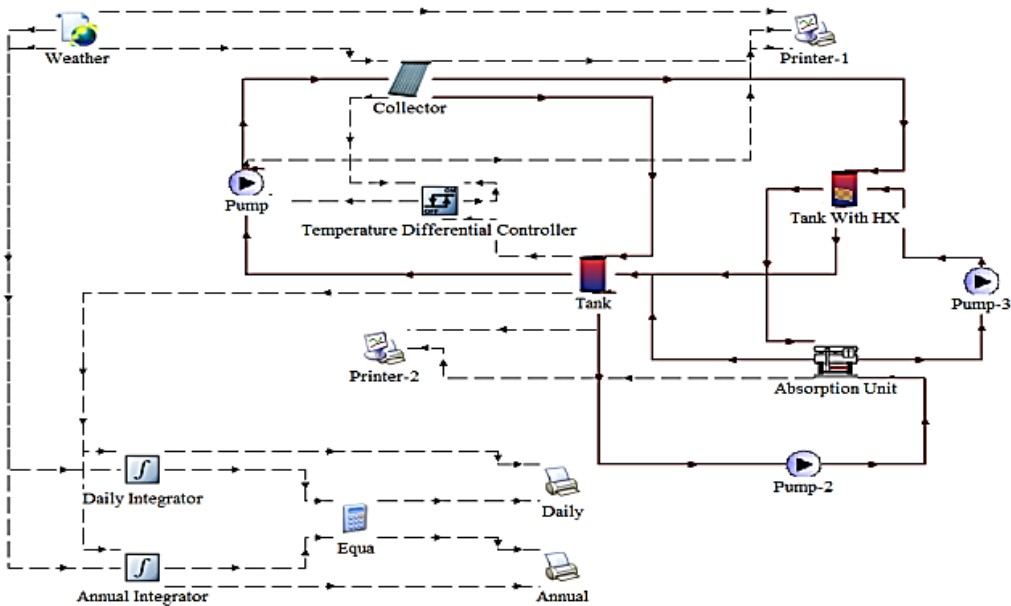

**Figure 3.** Simulation model of solar absorption refrigeration lyophilizer system.

Guangzhou is controlled by the subtropical high in July, with sunny days and the maximum solar altitude angle before and after the summer solstice. The maximum monthly total radiation appears in July. The solar radiation simulation result in July is shown in Figure 4.

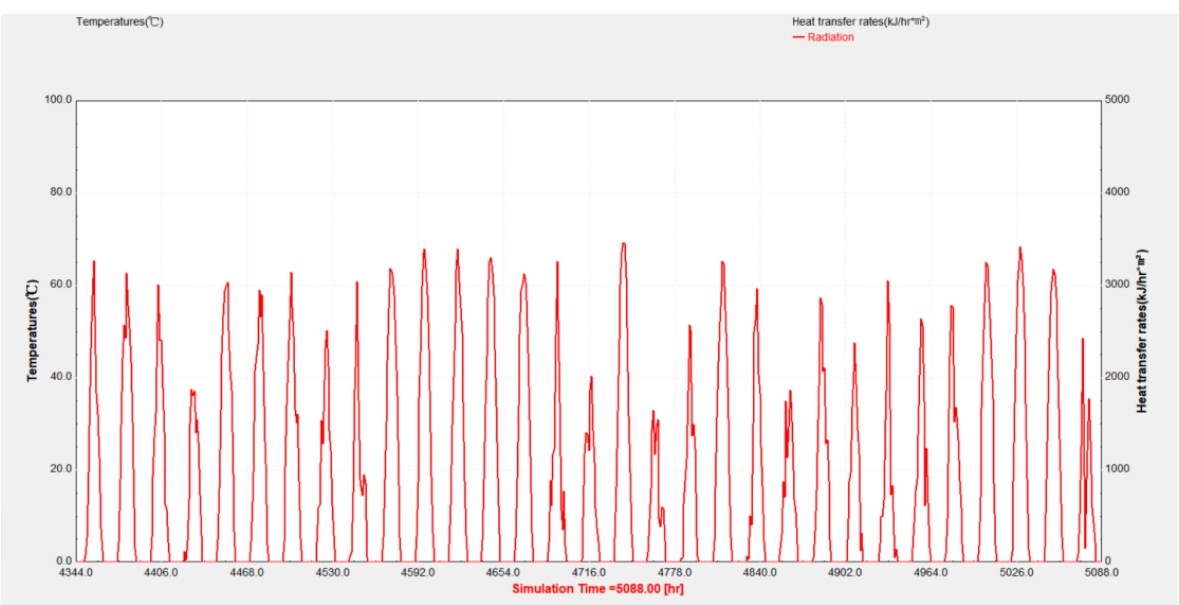

**Figure 4.** July solar radiation.

The annual solar radiation simulation result of this system is shown in Figure 5.

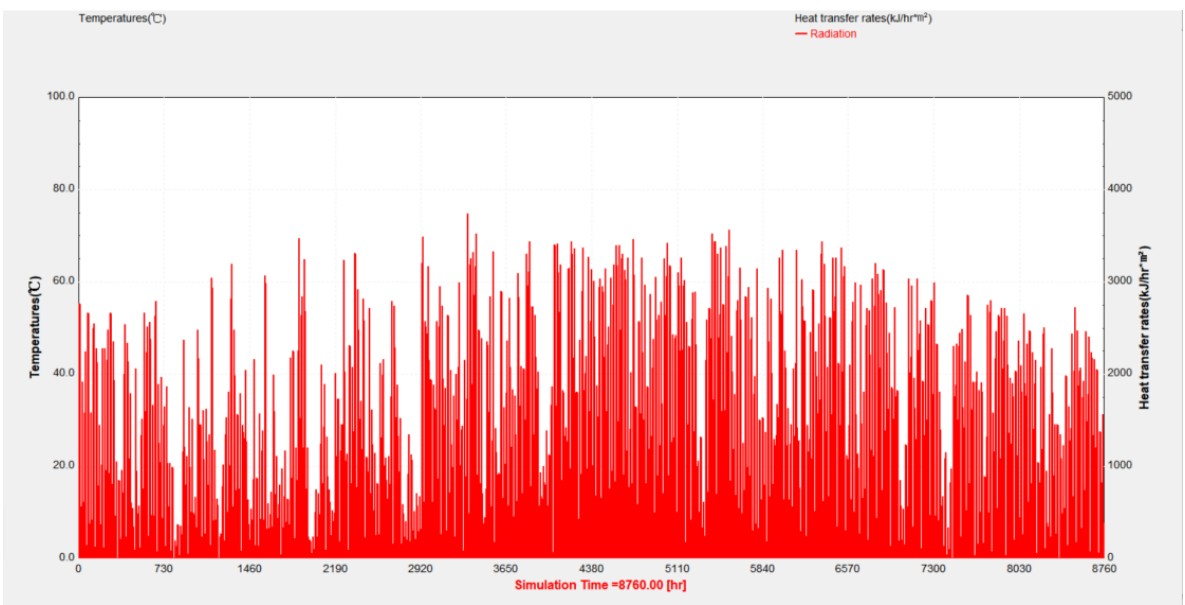

**Figure 5.** Annual solar radiation.

### 4.2. Energy Saving Calculation

According to the climate characteristics of a hot summer and warm winter in Guangzhou, this paper selects the data of a typical day, July 28, a typical month, July, and the whole year for calculation and analysis. The percentage of solar energy in the system load is explained by calculating the solar energy guarantee rate of the system. Solar guarantee rate is actually equal to the ratio of the useful energy of the collector and the load required by the system, and the solar guarantee rate Equation (2) is as follows:

$$f = \frac{Q_u}{L}, \tag{2}$$

In this equation, $f$ is the solar energy guarantee rate, %; $L$ is the load required by the system, W; $Q_u$ is the useful energy of collector output in unit time, W. Its Equation (3) is as follows:

$$Q_u = A_P I_T (\alpha\tau)_e - A_P U_L (T_P - T_a), \tag{3}$$

In this equation, $A_P$ is the collector area, m$^2$; $(\alpha\tau)_e$ is the product of the transmission ratio of the transparent cover plate and the absorption ratio of the heat absorber plate, without dimensional term; $U_L$ is the total loss coefficient of the collector, W/(m$^2 \cdot$ K); $T_P$ is the average temperature of the absorbing plate, °C; $T_a$ is the ambient temperature, °C; $I_T$ is the total radiation intensity on the inclined surface, W/m$^2$.

The total heat collection of the collector is calculated from the solar radiation output by the system, and the Formula (4) is as follows:

$$Q = J {\cdot} S {\cdot} A_P, \tag{4}$$

In this equation, $Q$ is the total collector heat, MJ; $J$ is the surface radiation of the collector, MJ/$\left(\text{h} {\cdot} \text{m}^2\right)$; $S$ is the collector duration, h; $A_P$ is the collector area, m$^2$.

The solar guarantee rate formula and simulation model of the data results can be calculated on July 28, July, and the annual of the solar guarantee rate, as shown in Table 3.

**Table 3.** Annual, typical month and typical day solar energy guarantee rate.

| Time | Total Solar Radiation Energy (MJ) | Available Energy (MJ) | System Load Requirement (MJ) | Solar Fraction (%) |
|---|---|---|---|---|
| July 28th | 9733.48 | 6927.98 | 9242.24 | 74.96 |
| July | 233,982.72 | 140,181.49 | 262,119.46 | 53.48 |
| Annual | 1,867,376.18 | 1,189,624.64 | 3,045,428.36 | 39.06 |

According to the chart, it can be concluded that, in July, the effective energy output of the collector was 140,181.49 MJ, the required load of the system was 262,119.46 MJ, the solar energy guarantee rate of the system was 53.48%, and the remaining 46.52% of the required load of the system was provided by the electric heater. On July 28, the solar energy guarantee rate reached 74.96%, and the annual solar energy guarantee rate was 39.06%. The effective energy output of the collector was the energy saving of the whole system, that is, 140,181.49 MJ of electric energy can be saved in July, which is equivalent to reducing the coal consumption by about 4790.9 kg; 6927.98 MJ of electric energy can be saved on a typical day, and the coal consumption can be reduced by about 236.8 kg; 1189624.18 MJ of electric energy can be saved throughout the year, and the coal consumption can be reduced by about 40,657.1 kg. The above data show that energy consumption can be saved only from the aspect of solar energy utilization, which plays a positive role in adjusting the energy structure. It can be seen more intuitively through Figure 6.

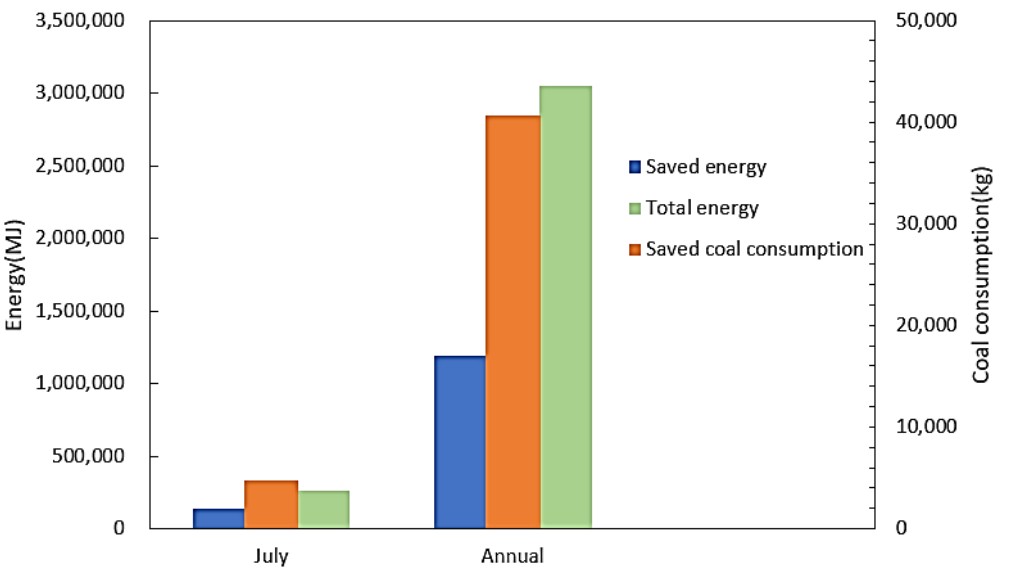

**Figure 6.** Typical annual and monthly energy consumption.

In addition, Zhou and Guo [25] proposed a system of a lyze-drying machine that uses solar energy absorption refrigeration in combination with buildings. Through the solar absorption refrigeration system, the cooling capacity is supplied to the lyophilizer and the building at the same time, so that the cooling capacity can be used step by step, which greatly saves energy consumption. The corresponding solar energy guarantee rate of the two systems is shown in Figure 7.

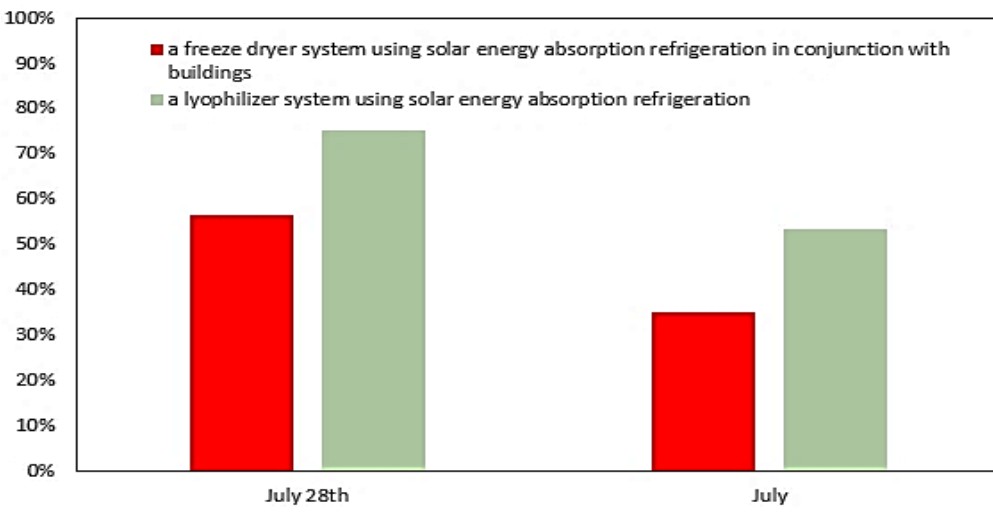

**Figure 7.** The corresponding solar energy guarantee rate.

*4.3. Result Analysis*

1.  The whole system is simulated by TRNSYS software, and the probability of the system is calculated. It can be seen from the calculation results that, compared with the freeze dryer using only electric heater as heat source, the freeze dryer system using solar absorption refrigeration can save energy significantly. On a typical day, the solar energy guarantee rate is 74.96%, which can save about 236.8 kg of coal. In a typical month, the solar energy guarantee rate is 53.48%, which can save about 4790.9 kg of coal. The annual solar energy guarantee rate is 39.06%, which can save about 40,657.1 kg of coal.

2.  The solar absorption refrigeration system can not only use the waste heat to defrost the cold trap, but can also combine with the building. The excess cooling capacity of the lyophilizer is used for building air conditioning refrigeration, which shows that the application prospect of the system is huge.

**5. Discussion and Analysis**

1.  Using this system, the freeze-dryer can not only stably absorb and store the heat discharged by each system during the operation of indoor air conditioning, but can also apply this heat to the defrosting system that needs to provide heat during the operation of the freeze-dryer, make full use of the system's waste heat, save the investment of cooling tower, reduce the energy consumption of the freeze dryer, and reduce the freeze-drying cost. In calculating the total energy consumption of the 20 m$^2$ vacuum freeze dryer, Shi and Lou [7] concluded that the refrigeration system accounts for 70% of the total energy consumption, and the sublimation electric heating system accounts for 17%. The freeze-drying cycle using the waste heat of the refrigeration system can theoretically save nearly 17% of the energy, with great energy-saving potential.

2.  As the main energy supply of the system, solar energy can greatly reduce the consumption of conventional energy and then play the role of energy conservation and environmental protection. In the heat collection system, the solar collector is used as the heat source of the system, and the vacuum tube collector is used to eliminate the convective heat loss and reduce the heat conduction loss. Vacuum tubes have large heat capacity and can obtain heat in a short time. In addition, vacuum tubes have excellent frost resistance and can be widely used in many fields. However, the initial investment of solar energy systems is generally high. As a new energy utilization technology, the economy of the whole system needs to be further analyzed.

3.   Both the system and Zhou's system adopt solar absorption refrigeration. The difference is that Zhou's system adds a heat exchange unit composed of waste heat exchanger, building an air conditioning heat exchanger and other components. The heat exchange unit is connected with the cold trap coil of the lyophilizer unit to replace the original refrigeration system of the lyophilizer unit, providing the required cooling capacity for the lyophilizer unit, and is connected with the water collector and water separator of the building air conditioning unit, The excess cooling capacity from the lyophilizer unit is used for the refrigeration of the building air conditioning unit. The cooling capacity can be used step by step, which greatly saves energy consumption. Compared with his system, the system adds a heat storage unit composed of a defrosting water storage tank and an air conditioning waste heat exchange mechanism. The defrosting water storage tank is connected with the condenser and absorber of the absorption refrigeration unit through the air conditioning waste heat recovery mechanism for recovering the waste heat of the condenser and absorber, and the defrosting water storage tank is connected with the solar heat collection unit for heat and mass exchange. The defrosting valve connected with the defrosting water storage tank can supply the heated defrosting water to the cold well for defrosting, so as to reduce energy consumption.

4.   Taking the process with evaporation temperature of $-38\,^{\circ}\mathrm{C}$ and load of $28.2 \times 10^6$ kJ/h as an example, Shi and Zhang [26] compare the energy consumption of compression refrigeration and absorption refrigeration. It is concluded that the power consumption of compression refrigeration is more than 12 times that of absorption refrigeration, and the advantages will become more prominent with the increase of scale and refrigeration depth. This shows that compared with compression refrigeration, absorption refrigeration has good economic benefits.

5.   The technical scheme of the system is suitable for any lyophilizer, especially for large lyophilizers. At the same time, the technical scheme of the system can not only be applied in a single machine, but also a set of heat storage systems can be shared by multiple lyophilizers to form a composite application of heat storage stations. The technical details of this multi system joint operation and its regional expansion need to be further analyzed.

**Author Contributions:** Conceptualization, H.Z. and Y.G.; methodology, H.Z.; software, H.Z.; validation, H.Z., Y.G. and Y.L.; formal analysis, H.Z.; investigation, H.Z.; resources, Y.G.; data curation, H.Z.; writing—original draft preparation, H.Z.; writing—review and editing, H.Z.; visualization, H.Z.; supervision, Y.G.; project administration, Y.G.; funding acquisition, Y.G. All authors have read and agreed to the published version of the manuscript.

**Funding:** This paper is supported by National Natural Science Foundation of China (Grant No. 51606116) and Major Scientific and Technological Research Projects of Shanghai Science and Technology Commission (Grant No. 19195810800).

**Conflicts of Interest:** The authors declare no conflict of interest.

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
