# Peer review of "Analysis of Energy Consumption of the Lyophilizer System Using Solar Absorption Refrigeration"

_sustainability, doi:10.3390/su132112063_

Round 1

Reviewer 1 Report

The general idea of this manuscript is interesting.  Design of a lyophilizer plant in Guangzhou as a system consisting  three units:  solar energy collection unit, absorption refrigeration unit and heat storage unit. The idea is not exactly new but the application is innovative. The system is modelled using TRNSYS software. The manuscript has some potential, although no experimental data are available. However theoretical only investigation requires much more accuracy and more elaborated comparisons:

  1. The system energy consumption shall be compared with classical solutions
  2. The solar energy gain is only one parameter, other components as electric energy consumptions shall also be considered. Especially fans for heat exchangers are very costly in energy.
  3. The tables with thermodynamic parameters in each characteristic point shall be included, also assumed efficiencies of all system elements shall be shown. This is required also for comparison with conventional systems to show the advantages in total energy consumption with conventional lyophiliser.

Reviewer 2 Report

The paper is a resubmitted manuscript. I have reviewed twice the original version of the paper. In this version I see that my recomendtions were included. Authors have explained all of my doubts. The paper looks better, and the quality of the paper increase. I do't have further objections and comments. The paper can be published

Author Response

Thank you very much for your comments and wish you a happy life.

Reviewer 3 Report

I believe that the revised document is not complete, as it lacks the conclusions, which it previously had. 

I believe that there is no understanding between my comments and the authors and that is why we are still in this process. I find myself unable to explain myself better.

When I say to represent in Figure 6 the annual data it is represented by months, weeks or as the authors have the data, but it is not consistent to evaluate a document with data of one month and one day. If this is the only data available, which it seems it is not (table 1), the paper is strongly weakened by data supporting the results. Furthermore, the data should be commented on in the text.

The answer to point 3 should be integrated into the discussion. The discussion section is a comparison of the results of the authors of this paper with those obtained by other published papers. This section is very important because it highlights the merits of the results obtained and helps to draw more valuable conclusions. Therefore, it was pointed out that comparison was not enough, more comparisons are needed. This section cannot be a summary of the results.

Round 2

Reviewer 1 Report

I can not fully agree with your respond. This sentence "Conventional compressor refrigeration equipment uses 100% electric power or medium pressure steam as the driving energy, while the waste heat absorption refrigeration system uses only about 10% electric energy as the driving energy, and the other 90% of the driving energy directly comes from heat energy" is to general and in most cases not true. The power consumption for fans in the absorption system is usually much higher. Besides the table with relevant parameters shall be included in the manuscript not only in the answer. However I am nearly satisfied with the response.

Author Response

Thank you for your suggestion. In order to prevent misleading readers, I deleted that sentence and directly compared compression refrigeration with absorption refrigeration by taking the refrigeration process with evaporation temperature of -38℃ and load of 28.2x106 kJ/h as an example. In addition, thank you very much for your reminder. The table with parameters has been inserted into the manuscript.

Reviewer 3 Report

Changes accepted

Author Response

Thank you for your comments and suggestions. I have gained a lot from it. It is my honor to meet you. I will continue to work hard in my future study and life. Thank you again.

This manuscript is a resubmission of an earlier submission. The following is a list of the peer review reports and author responses from that submission.

Round 1

Reviewer 1 Report

There should be case study relations of lyophilizer system 
using solar energy absorption refrigeration and their workability. This should be presented to show cases of energy saved or loss from the process. Using this on a long term basis, what will be the cost effectiveness rather than just over a period in the month of July. What is the idea here when you wrote about "July" and "28th July"? Why this? Explain the reason behind these dates 28th July and July. Why July and not other months of the year? Give reasons for this choice? 

Reviewer 2 Report

The manuscript presents an idea of reducing the energy consumption in a lyophilizer system using solar energy absorption refrigeration. The solar energy is used to defrost the cold trap of the lyophilizer. The work is strictly theoretical, although the application in r a lyophilizer factory in Guangzhou China is mentioned. The simulation is done using TRNSYS. The system has been granted a Chinese patent.

Wekanesses:

  1. No real experimental validation of the model, even foreseen application is only simulated.
  2. The descritpion of the model is not satisfactory, it requires a table with given pressure, temperature and energy flow in all devices. The devices efficiencies and energy losses shall be included. 
  3. Even the refrigerant used in adsorption system is not shown.
  4. The electric power for all equipment (fans, pumps) shall be considered and shown in a table.
  5. Theoretical work requires also simulation results for several different operating parameters probably depending on the ambient conditions and system load. Once there is a model this is an easy task to do.
  6. The comparison with compression system would be an added value. The comparison shall include also costs.

The manuscript in my opinion needs serious rewriting before accepting for publication.

Reviewer 3 Report

abstract should be modified. It should shortly describe the problem, method applied and results obtained using some quantitative values.

what are the novelties? what are the benefits resulting from this paper?

the paper present result obtained for one day of system operation, and for one month. In my opinion it is not enough. It presents the operation of the system during summer. Authors should also present the system operation at other times of the year

are there any experimental validation for these results?

simulation procedure is poorly desribed

quality of figures 1-3 are very poor, e.g. lines described on the figure 3 legend are not visible

Reviewer 4 Report

The authors propose a vacuum lyophilizer model, which has been patented. They perform simulation on data from lyophilizer factory in guangzhou China, using TRNSYS freeze dryer workshop software. The system is interesting and well described.  The authors already have another publication applied to solar absorption cooling in combination with buildings (ref. 9).

The main inconvenience I observe to support the good performance of the model is that only one month and one day of data is used, which is insufficient. The authors should elaborate the paper with more results. In addition, it needs a discussion section to introduce readers to the benefits of the proposed model in relation to existing and published models.

  In order to improve the document, the following changes are recommended:

  • The first three sections can be incorporated into a single introductory section. In this way, readers are better oriented towards the objective of the work. The only requirement is to redirect the connections between the three sections so that it is fluid.
  • In section 4 the idea introduced in the previous sections is repeated, I think it should be rewritten highlighting the important parts to define the model and merged with section 5 of the document. It is important to condense the information in the sections, each section should provide valuable information for the readers, so that they do not get lost in technicalities and the document becomes more relevant as it can be more communicative and fluid.
  • Figures 1, 2 and 3 are of poor quality, they look pixelated. Check and improve their quality.
  • The units of measurement should be added to figure 3. The temperature scale and heat transfer rates should be adjusted to the maximum values in order to look better and to be able to quickly identify the maximum values. Wouldn't it be better to represent the simulation time scale in terms of months rather than hours? At the top there are acronyms that have not been defined and that are difficult to identify in the graph. I don't quite understand the figure, the simulation is based on data collected from the solar radiation of the area or those that are going to be applied for the verification of the model. Why is the minimum value always -300 in temperature? I think this graph should be better explained in the document. In the next section it only uses one month and one day of the month for the calculations, so I understand figure 3 even less.
  • -In section "6.2. Calculation of energy savings" the authors say: "the data for July and July 28th have been selected for the calculation and analysis in this work" as this is a simulation and I assume that the solar radiation data for the test area are known, because not all the annual data or a daily average for the last 5 years, for example, have been taken into account. Data from one month or one day are not meaningful to be able to evaluate the performance of the model. Solar radiation does not have a homogeneous distribution and its capture depends to a large extent on the angle of inclination of the panel according to geographical location and its efficiency. These data have not been discussed in the document and I believe that they are important, as table 1 gives values of Available Energy for the total energy of solar radiation.
  • In table 1 there is no space between the parameters and their units of measurement, e.g. Available Energy(MJ) instead of Available Energy (MJ).
  • In section "6.2. Energy saving calculation" the authors say "In this equation f is the solar energy guarantee rate, %; L is the load required by the system, W; Qu is the useful energy of collector output in unit time, W. Its equation (3) as follows:", what is meant by W.
  • Figure 4 of the paper is the same as Figure 6 of the paper: Zhou, Z., Guo, Y., & Lin, Y. (2021). Energy-saving evaluation of a solar integrated vacuum freeze-dryer and building air conditioning system. Energy Exploration & Exploitation, 39(2), 608-619. This time it has less quality. This figure should be modified to incorporate new and more data.
  • The paper does not have a discussion section to introduce readers to the advantages of the proposed model in relation to existing published models. This would allow readers to assess the significance of the findings. And this section could help to improve the conclusions.
  • The conclusions should go beyond a compilation of results, they should indicate the merits of the model, but for this it needs to have more results and a good comparison with other models. This section should be rewritten once the sections related to the results and discussion have been modified.
  • In the conclusions section, the authors say "In the total energy consumption of 20㎡ vacuum freeze dryer, the sublimation electric heating system accounts for 17%. Theoretically, the freeze-drying cycle using the waste heat of the refrigeration system can save nearly 17% of the energy, which has great energy saving potential". But they do not present this data, the calculations have been made for 75m2.
  • The references should be updated to more up to date ones.
  • After passing the document through the anti-plagiarism software, the percentage is shown as 27%. There is a 21% overlap with another paper by two of the authors, reference 9: Zhou, Z., Guo, Y., & Lin, Y. (2021). Energy-saving evaluation of a solar integrated vacuum freeze-dryer and building air conditioning system. Energy Exploration & Exploitation, 39(2), 608-619. Authors should highlight the differences between the two papers. I will attach the file.

Round 2

Reviewer 2 Report

Most of my concerns were not answered. Lack of any experimental results, verification, lack of comparison with standard compression systems. Author states that they have plans to work on those issues in future work. Then the manuscript would be good for publication. 

Author Response

Thank you for your comments. Because the manuscript proposes a new system and has not been verified by experiments, there are no corresponding experimental results at present. In addition, this paper proposes a theoretical model and does not select the specific model of each component, so it is unable to compare the energy consumption and economy with the compressed system. I hope to get your understanding. In addition, I would like to explain the reason for choosing the absorption refrigerator: compared with the compression refrigeration, which needs to consume high-grade electric energy, the absorption refrigeration has low requirements for the consumption of heat energy, and can make good use of the heat energy converted by solar energy. Moreover, the absorption refrigerator has the advantages of simple structure, convenient installation and low requirements for buildings, so the investment outside the equipment (materials, civil engineering and construction costs) is relatively low.

I'm sorry I didn't meet your satisfaction, but thank you very much for your comments and suggestions. I will study hard in my future work. Thank you for your work.

Reviewer 3 Report

Authors have answered to all questions and provided explanations as the Reviewer suggested

Author Response

Thank you again for your comments and suggestions, I will study hard in my future work. And thank you for your hard work.

Reviewer 4 Report

The changes made have improved the document, but there are still issues that have not been satisfactorily resolved or raise doubts in my mind, some of the changes are not clear in the document where the changes have been mixed with what was there before, which makes typos. I recommend the authors to mark in another colour what is new and delete what does not correspond, so that the changes can be better understood.

There are issues that have not been resolved such as adding a discussion section among others. I list them again so that you can take them into account:

  1. Check the entire document for typographical errors, many of which are a consequence of the changes made. Delete what does not correspond to the revised version and mark in another colour what has been changed.
  2. The current figure 5, now there are two figures represented, the old one and the new one, I understand that the old one will not be in the final document. It should be revised, as the rest of the figures have been replaced by the new ones with higher quality and the graphs with the units on the axes. I don't quite understand the figure, the simulation is based on data collected from the solar radiation of the area or those that are going to be applied for the verification of the model. I think this graph should be better explained in the document.
  3. In table 1 there is no space between the parameters and their units of measurement.

Total solar radiation energy(MJ)             Available Energy(MJ)  

System Load Requirement(MJ)               Solar Fraction(%)

It should say:                                                                 

Total solar radiation energy (MJ)            Available Energy (MJ)

System Load Requirement (MJ)              Solar Fraction (%)

  1. In the figures “Figure 4. Total electric energy, saved electric energy and saved coal consumption of the system in typical day and month” and “Figure 56. The corresponding solar energy guarantee rate” as well as the text where it is commented, annual data should be added, which can be more significant. In the previous review I already commented on the importance of working with a significant amount of data, although the month of July may be the month of highest consumption, it is also one of the months with the highest solar capture. The annual data can be used to obtain more relevant data.
  2. The paper does not have a discussion section to introduce readers to the advantages of the proposed model in relation to existing published models. This would allow readers to assess the significance of the findings. The "Result analysis" section cannot replace the discussion section.
  3. In the conclusions section, the authors say "In The system makes use of the total energy heat discharged during the operation of the absorption refrigeration unit, makes the absorption refrigeration unit achieve the effect of waste reuse, and saves the energy consumption of 20㎡ vacuum freeze dryer, the sublimation electric heating system accounts for 17%. Theoretically, the freeze-drying cycle using the waste heat of the the". But they do not present this data (20㎡ vacuum freeze dryer), the calculations have been made for 75m2.
  4. After passing the document through the anti-plagiarism software, the percentage is shown as 27%. There is a 21% overlap with another paper by two of the authors, reference 9: Zhou, Z., Guo, Y., & Lin, Y. (2021). Energy-saving evaluation of a solar integrated vacuum freeze-dryer and building air conditioning system. Energy Exploration & Exploitation, 39(2), 608-619. Authors should highlight the differences between the two papers. I will attach the file.

Round 3

Reviewer 2 Report

I must maintain my final conclusion from the review of the previous version. The manuscript has not been improved sufficiently. In addition, the use of solar energy alone does not guarantee savings in the real case, because when using absorption cycles, efficient cooling is necessary to lower the intermediate temperature of the absorber and the condenser, so a high-power fan for the cooling tower must be used and its energy consumption included in the calculations.

Reviewer 4 Report

The changes made have improved the document, but there are still issues that have not been satisfactorily resolved. I list them again so that you can take them into account:

  1. In Figure 6 and in the text where it is commented should be added, in both, annual data (Fig. 6 and text). In the previous review, I already commented on the importance of working with a significant amount of data. With annual data, more relevant data can be obtained.
  2. The authors should make a more exhaustive discussion, a comparison is not enough.
  3. The authors should highlight the differences between the two papers: reference 9 and this paper in the discussion.